# The Effect of Hydrogen on Martensite Transformations and the State of Hydrogen Atoms in Binary TiNi-Based Alloy with Different Grain Sizes

**DOI:** 10.3390/ma12233956

**Published:** 2019-11-28

**Authors:** Anatoly Baturin, Aleksandr Lotkov, Victor Grishkov, Ivan Rodionov, Yerzhan Kabdylkakov, Victor Kudiiarov

**Affiliations:** 1Institute of Strength Physics and Materials Science of the Siberian Branch of the Russian Academy of Sciences, Tomsk 634055, Russia; abat@ispms.tsc.ru (A.B.); grish@ispms.tsc.ru (V.G.); rodionov231@mail.ru (I.R.); 2Division for Experimental Physics, School of Nuclear Science and Engineering, National Research Tomsk Polytechnic University, Tomsk 634050, Russia; kabdylkakov96@mail.ru (Y.K.); viktor.kudiiarov@gmail.com (V.K.)

**Keywords:** binary TiNi-based alloy, hydrogen, martensitic transformations, electrical resistivity, thermal desorption spectroscopy

## Abstract

The analysis presented here shows that in B2-phase of Ti_49.1_Ni_50.9_ (at%) alloy, hydrogenation with further aging at room temperature decreases the temperatures of martensite transformations and then causes their suppression, due to hydrogen diffusion from the surface layer of specimens deep into its bulk. When hydrogen is charged, it first suppresses the transformations B2↔B19′ and R↔B19′ in the surface layer, and when its distribution over the volume becomes uniform, such transformations are suppressed throughout the material. The kinetics of hydrogen redistribution is determined by the hydrogen diffusion coefficient D_H_, which depends on the grain size. In nanocrystalline Ti_49.1_Ni_50.9_ (at%) specimens, D_H_ is three times greater than its value in coarse-grained ones, which is likely due to the larger free volume and larger contribution of hydrogen diffusion along grain boundaries in the nanocrystalline material. According to thermal desorption spectroscopy, two states of hydrogen atoms with low and high activation energies of desorption exist in freshly hydrogenated Ti_49.1_Ni_50.9_ (at%) alloy irrespective of the grain size. On aging at room temperature, the low-energy states disappear entirely. Estimates by the Kissinger method are presented for the binding energy of hydrogen in the two states, and the nature of these states in binary hydrogenated TiNi-based alloys is discussed.

## 1. Introduction

Near-equiatomic TiNi alloys with thermoelastic martensite transformations, shape memory effect (SME), and superelasticity (SE) are used in engineering and medicine [1]. However, when applied to medical products (e.g., implants and dental devices, which are in long-term contact with hydrogen-containing saline and human tissue,), alloys such as these reveal hydrogen embrittlement and influence of absorbed hydrogen on their martensite transformations and functional properties—in particular, SME and SE [2,3,4]. It is this problem that stimulates the research in the interaction of hydrogen with near-equiatomic TiNi [5,6,7,8,9,10,11,12,13,14,15,16,17,18,19,20,21,22,23,24,25,26,27,28,29,30,31,32,33,34,35].

By now, it is known that hydrogenation, whether gas phase or electrolytic, suppresses the B2↔B19′ or R↔B19′ transformations in B2-phase binary TiNi-based alloys [5,6,7,8]. In addition, according to resistivity measurements [7], it extends the temperature range of premartensite effects. During electrolytic hydrogenation, hydrogen atoms in specimens are first concentrated in its near-surface layer and are then redistributed by diffusion into its bulk. The kinetics of such redistribution is determined by the hydrogen diffusion coefficient D_H_, which depends on the structure and phase state of alloys. From hydrogen profiles [8], it follows that the martensite phase of TiNi-based alloys features a lower D_H_ value than its austenite phase. Studies on other metal materials [9] show that this coefficient also depends on the grain size and that its effective value can be lower or higher than the bulk diffusion coefficient, depending on what type of grain boundaries (low-or high-angle) dominates in a material. Such measurements for TiNi-based alloys have not been made so far.

For understanding the influence of hydrogen on the properties of TiNi-based alloys, we should know the state of hydrogen atoms in the materials, e.g., whether they are interstitial or trapped by defects like vacancies, dislocations, grain boundaries, etc. The state of hydrogen atoms in TiNi-based alloys can be judged from their thermal desorption spectra [10,11,12,13,14,15,16,17,18,19,20,21,22]. According to the majority of studies, the thermal desorption spectrum of TiNi-based alloys comprises two overlapping peaks which correspond to low and high temperature, respectively. The low-temperature peak is usually related to mobile hydrogen atoms at hydride interstices, and the high-temperature peak to those trapped by defects. Moreover, after aging for 240 h, only one high-temperature peak remains in the hydrogen desorption spectrum of Ti_45_Ni_55_ (wt%) alloy [22], but the cause of this is unclear. To correctly judge what lies behind the peaks in thermal desorption spectra, we need to know the binding energy of hydrogen with surrounding lattice atoms for which thermal desorption should be measured for TiNi-based alloys at different heating rates. Up to now, no such measurements have been performed.

Here we analyze the influence of hydrogen atoms and their diffusion on martensite transformations in coarse-grained and nanocrystalline B2-phase Ti_49.1_Ni_50.9_ (at%) alloy after electrolytic hydrogenation with further aging at room temperature.

## 2. Materials and Methods

The test alloy was Ti_49.1_Ni_50.9_ at% (hereinafter Ti_49.1_Ni_50.9_) supplied as wires of diameter 1.1 mm (MATEK-SMA, Ltd., Moscow, Russia). The as-received wire had a nanocrystalline structure with an average grain/subgrain size of 85 nm. Its coarse-grained structure with near-equiaxed grains of average size 10 µm was obtained by annealing at 923 K for 30 min with further water quenching. The microstructure of both types of specimens was analyzed elsewhere [7].

The Ti_49.1_Ni_50.9_ specimens were electrolytically hydrogenated in a 0.9% NaCl solution at a current density of 20 A/m^2^ for 3 h. The hydrogen concentration was measured on a LECO RHEN 602 gas analyzer (LECO, St. Joseph, MI, USA). The initial hydrogen content, i.e., immediately after hydrogenation, was 500 ± 50 ppm and 260 ± 50 ppm for nanocrystalline and coarse-grained Ti_49.1_Ni_50.9_ (at%), respectively.

The effect of aging at room temperature on the sequence and temperature of martensite transformations was analyzed from temperature dependences of resistivity ρ(T) measured by the four-probe potential method at 83–333 K.

The thermal desorption of hydrogen was measured on a Gas Reaction Controller automated complex with an RGA100 quadrupole mass spectrometer (Stanford Research Systems, Inc. Sunnyvale, CA, USA). For the measurements, the specimens placed in the vacuum chamber with a residual pressure no greater than 10^−4^ Pa were heated to 300–1173 K at a constant rate of 4, 6, and 8 K/min. The binding energy of hydrogen atoms with surrounding lattice atoms was determined by the Kissinger method [23].

## 3. Results and Discussion

Figure 1 shows the temperature dependences of resistivity ρ(*T*) for nanocrystalline and coarse-grained Ti_49.1_Ni_50.9_ immediately after hydrogenation and after aging at room temperature. Immediately after hydrogenation, the nanocrystalline and coarse-grained specimens preserve the sequence of their respective phase transformations B2↔R↔B19′ and B2↔B19′, where B2 is high-temperature austenite, R and B19′ are the rhombohedral and monoclinic martensite phases, respectively. Pretransition phenomena caused the small increase in the resistivity on cooling, before the onset of B2→B19′ transformation—both in the initial state [7] and after hydrogenation with aging at room temperature for 1 h (Figure 1b). It is possible that the growth of electrical resistivity at cooling of hydrogenated and aged specimens can also be caused by the formation of “strain glass state”, which was discussed for Ti_50−X_Ni_50+X_ (at%), Ti_50_Ni_44.5_Fe_5.5_ (at%) and Ti_49_Ni_50−x_Pd_X_ (at%) alloys [36,37,38,39]. However, experimental confirmation of this assumption for hydrogenated Ti_49.1_Ni_50.9_ (at%) alloy is absent at present. Table 1 shows the martensite transformation temperatures in nanocrystalline and coarse-grained Ti_49.1_Ni_50.9_ before and after hydrogenation with aging for 3 h and 1 h, respectively. It is seen from Table 1 that the coarse-grained specimens have almost the same temperatures of B2↔B19′ transformations before and after hydrogenation. In the nanocrystalline specimens, the temperatures T_R_, M_s_, A_s_, and A_f_ also remain unchanged after hydrogenation with aging for 3 h. However, the finish temperature of R→B19′ transformation in these specimens is noticeably lower than its value in the initial specimens (165 K and 182 K, respectively). Obviously, during aging for 3 h, the diffusion of hydrogen from the surface layer of the hydrogenated nanocrystalline specimens decreases the temperature of R→B19′ transformation in the underlying layer. As the aging time is increased, both types of specimens reveal qualitatively similar changes in their ρ(T) dependences on cooling and heating from 83 K to 323 K.

After long-term aging at room temperature, the ρ(T) dependences of the hydrogenated specimens reveal only a B2→R transformation and premartensite effects: An increase in ρ without its steep decrease characteristic of R→B19′ and B2→B19′ [24]. Increasing the aging time greatly decreases the hysteresis of the ρ(T) dependences of hydrogenated Ti_49.1_Ni_50.9_ on cooling and heating in the temperature range of martensite transformations (Figure 1). Most likely, the small hysteresis of ρ(T) in the long-aged nanocrystalline and coarse-grained specimens owes to B19′ transformation in their small near-central volumes with a lower hydrogen content compared to their intermediate layers. Thus, the hydrogen distribution in the nanocrystalline and coarse-grained specimens remains inhomogeneous after aging for 789 h and 650 h, respectively.

As has been shown [8,14,22,25], electrolytic hydrogenation of TiNi-based alloys in saline produces a tetragonal TiNiH structure in its surface layer, and further aging at room temperature provides hydrogen diffusion deep into its bulk. The hydride structure in the surface layer can either survive up to strong hydrogen depletion [25], or dissociate completely [14,22]. The diffusion of hydrogen atoms is directed only from the surface of hydrogenated specimens deep into its bulk, because the oxide barrier precludes their escape from the material at room temperature. During the aging time Δ*t*, the hydrogen diffusion distance measures Δ*r* = (2*D*_H_Δ*t*)^1/2^. It is supposed that such diffusion allows hydrogen atoms to gradually fill octahedral B2 interstices whose immediate surroundings contain four Ti atoms and two Ni atoms [26]. If the average hydrogen concentration in a given layer is higher than 100 ppm, the transformation to B19′ in this layer is suppressed [27]. In our specimens, the average hydrogen concentration was far greater than the above critical value.

As hydrogen atoms diffuse deeper, filling octahedral B2 interstices, more and more material volume ceases to experience B19′ transformations, and simultaneously, a wider temperature range is provided for premartensite B2-phase. It is these premartensite effects that increase the resistivity of Ti_49.1_Ni_50.9_ at 83 K and decrease the hysteresis on its ρ(T) curves. The hydrogen redistribution in TiNi-based alloys which experience martensite transformation can be outlined as follows. Immediately after hydrogenation, almost all hydrogen atoms are concentrated in a surface layer ~20–30 µm thick, and hence, almost all specimen volume is free of hydrogen and capable for martensite transformations like before hydrogenation. With time (*t*), progressively deeper layers are penetrated by hydrogen, and thus, prevented from martensite transformations, and when its atoms reach the specimen center (*t*_sat_), the entire specimen volume becomes incapable for such transformations.

The ρ(T) curves of hydrogenated specimens show a decrease in the maximum resistivity difference Δρ on heating and cooling at martensite transformation temperatures (Figure 2, insert) [28], due to the smaller material volume involved in such transformations. The volume of suppressed martensite transformations can be estimated as υ = (1 − Δρ/Δρ_0_), where Δρ_0_ is the initial value of Δρ before hydrogenation. Thus, we have Δρ/Δρ_0_ = 1 and υ = 0 before hydrogenation and Δρ/Δρ_0_ = 0 and υ = 1 after aging.

From υ we can also estimate the time *t*_sat_ it takes for hydrogen atoms to diffuse over the specimen volume up to its center, i.e., to the point υ = 1 at which the transformation to B19′ is suppressed throughout the material. Figure 2 shows υ = (1 − Δρ/Δρ_0_) as a function of the aging time for hydrogenated Ti_49.1_Ni_50.9_.

As can be seen from Figure 2, the nanocrystalline Ti_49.1_Ni_50.9_ specimens reach υ = 1 at *t*_sat_ = 430 ± 10 h, and the coarse-grained ones fail to do it on the time interval studied (up to 654 h). From extrapolation of υ(*t*) to υ = 1, this value in the coarse-grained material is reached at *t*_sat._ = 1300 ± 20 h.

The hydrogen diffusion coefficient can be estimated as *D*_H_ = *r*^2^/2*t*_sat_ [29]. Its values for coarse-grained and nanocrystalline Ti_49.1_Ni_50.9_ are presented in Table 2.

As can be seen from Table 2, our estimates of *D*_H_ in coarse-grained Ti_49.1_Ni_50.9_ agree well with the values available. The data also suggest that the hydrogen diffusion in the nanocrystalline specimens is much faster than in the coarse-grained ones. Reasoning that the average grain sizes in the two types of specimens differ, it is likely the grain boundaries which accelerate the hydrogen diffusion in nanocrystalline Ti_49.1_Ni_50.9_. However, the hydrogen diffusion coefficient *D*_H_ depends strongly on what type of grain boundaries dominates in a material [9]. Low-angle boundaries, associated with dislocations, are normally a trap for hydrogen atoms and are, thus, a barrier to their diffusion [9]. High-angle boundaries feature an increased free volume favorable for hydrogen diffusion along with them. According to transmission electron microscopy [7], our nanocrystalline Ti_49.1_Ni_50.9_ specimens are dominated by equiaxed grains with high-angle boundaries, and this is likely the cause for fast hydrogen diffusion.

The effect of longer aging on the hydrogen distribution can be qualitatively analyzed by comparing the data in Figure 1 and Figure 3, which shows the ρ(T) dependences of the hydrogenated specimens of diameter 1.1 mm and 0.49 mm after aging for 115 days (2760 h). As can be seen, increasing the aging time from 789 h (Figure 1) to 2760 h (Figure 3) causes a slight decrease in the ρ(T) hysteresis on cooling and heating for the specimens of diameter 1.1 mm. At the same time, for the specimens of diameter 0.49 mm, this hysteresis decreases substantially (≤3 deg, which compares with the experimental setup capacity). The results suggest that after aging for 2760 h, the hydrogen distribution in the specimens of diameter 0.49 mm is much more homogeneous compared to the specimens of diameter 1.1 mm. Besides, in the nanocrystalline material, the time it takes for hydrogen atoms to almost uniformly distributed over its cross-section is much longer than *t*_sat_ (430 h).

For a better understanding the influence of hydrogen on martensite transformations in TiNi-based alloys, we should know the state of hydrogen atoms in the material, because this influence can differ depending on whether their preferential sites are interstices or lattice defects [27,28]. The state of hydrogen atoms in Ti_49.1_Ni_50.9_ was analyzed by thermal desorption spectroscopy of its freshly hydrogenated coarse-grained and nanocrystalline specimens having a B2 structure at room temperature. The heating rate was 6 K/min. The time between the end of hydrogenation and the beginning of desorption measurements was 3 h. Figure 4 shows the thermal desorption spectra of hydrogen for freshly hydrogenated Ti_49.1_Ni_50.9_.

It is seen from Figure 4 that the thermal desorption spectra for both types of freshly hydrogenated Ti_49.1_Ni_50.9_ are qualitatively similar, showing two overlapping peaks and general consistency with available data [10,11,12,13,14,15,16,17,18,19,20,21,22]. For better interpretation of such spectra in terms of several hydrogen atom states, they should be separated into components described by Gaussians [32]. In our case, the spectra are separated into two Gaussians (Figure 4). From the relative area beneath each Gaussian, we can judge the relative fraction of hydrogen atoms in a given state. Table 3 presents the relative areas *S*_1_, *S*_2_ beneath two Gaussians in the thermal desorption spectra of Ti_49.1_Ni_50.9_.

It is seen from Table 3 that for coarse-grained specimens, the relative fraction of hydrogen atoms in low-temperature states (*S*_1_) is 6% higher than that in high-temperature states (*S*_2_), and for nanocrystalline specimens, their relative fraction in high-temperature states is 10% higher than the other.

The low-temperature peak is related, as a rule, to mobile hydrogen atoms which occupy TiNiH lattice interstices. These are octahedral interstices with four Ti atoms and two Ni atoms in their immediate surroundings [26]. The high-temperature peak is often related to hydrogen atoms trapped by defects like vacancies and their complexes, dislocations, grain boundaries. In nanocrystalline metal systems, compared to coarse-grained ones, the density of such defects is higher [33], and this qualitatively agrees with the second peak prevailing in the thermal desorption spectrum of nanocrystalline Ti_49.1_Ni_50.9_. However, there are strong grounds to doubt that the high-temperature peak owes solely to hydrogen localization in defect regions. First, the TiNiH structure provides two crystallographically nonequivalent octahedral sites H_1_ and H_2_ for hydrogen atoms [34]. The energy-beneficial sites H_1_ are occupied by hydrogen atoms when their concentration is no greater than 0.5 atoms per metal atom (Ti, Ni). At higher concentrations, they occupy the sites H_2_ which are two times smaller in number than H_1_ [34], and at which the energy of hydrogen atoms should be higher. It can, thus, be expected that at high hydrogen concentrations, the thermal desorption spectrum of TiNiH will comprise two peaks, which does occur when almost all hydrogen atoms in freshly hydrogenated binary TiNi-based alloys are concentrated in their thin surface layer [14,22]. Hence, even without accounting for crystalline defects, hydrogenated TiNi-based alloys can show two peaks in their thermal desorption spectra. There is no doubt that the first peak is associated with TiNiH in hydrogenated Ni-rich TiNi-based alloys. First, as has been shown [35], the larger the number of hydrides formed in surface layers of Ti_49.15_Ni_50.85_ (at%) samples, the higher the intensity of the first peak. Second, if we heat a freshly hydrogenated specimen to 473 K, which corresponds to the temperature of the first peak, and then cool it to room temperature, no hydride reflection will be found on its X-ray pattern.

However, in view of available data [40], hydrogenated TiNi alloys with a higher Ti content may have another cause for the low-temperature peak of thermal desorption on heating. According to the data [40], no TiNiH hydride was detected in B19′-phase Ti_50.5_Ni_49.5_ (at%) alloy after hydrogenation. Instead, the material revealed a hydrogen-rich “H-strained” B19′ phase which transformed on heating to a B2 phase (hydrogen solid solution) with further hydrogen desorption from the B2 phase. By and large, the temperature intervals of hydrogen-rich “H-strained” B19′ transformation and B2 hydrogen desorption are close to the low- and high-temperature peaks in Figure 4, but any other conclusions need respective thermal desorption data which are not presented in the cited paper [40].

To clarify the nature of the second (high-temperature) peak in thermal desorption spectra, we analyzed how the state of hydrogen atoms in Ti_49.1_Ni_50.9_ changes after five-month aging, which is sufficient for hydrogen redistribution from the specimen surface to the bulk. The Ti_49.1_Ni_50.9_ specimens at room temperature had the same structure and phase state as the freshly hydrogenated ones. Figure 5 shows the thermal desorption spectra of hydrogen for freshly hydrogenated Ti_49.1_Ni_50.9_, and for its specimens after five-month aging.

It is significant that after long-term aging at room temperature, the low-temperature peak in the spectra of both types of Ti_49.1_Ni_50.9_ disappears and remains only the high-temperature peak whose intensity is much higher than that in the initial specimens. For the coarse-grained specimens, the temperature of this peak after aging is 19 K lower than its value after hydrogenation, and for the nanocrystalline specimens, it is 28 K higher. The temperature difference between the peaks owes to different binding energies of hydrogen atoms and surrounding B2 lattice atoms in the two states. However, to quantitatively estimate the binding energy of hydrogen and surrounding atoms, we should analyze the thermal desorption of hydrogen in hydrogenated Ti_49.1_Ni_50.9_ at different heating rates. Our measurements at a heating rate of 4, 6, and 8 K/min show that increasing the heating rate of the specimens increases the peak temperature in their thermal desorption spectra (Figure 6).

From the shift of peak temperatures, we determined the binding energy of hydrogen and surrounding B2 lattice atoms in nanocrystalline and coarse-grained specimens by the Kissinger method [23]. According to this method, the binding energy E is equal to the product of the gas constant R and the angular coefficient of the dependence of $\ln(\beta /T_{m}^{2})$ on $1/T_{m}$ (Figure 7), where β is the heating rate of a material and *T*_m_ is the peak temperature in its thermal desorption spectrum.

For nanocrystalline Ti_49.1_Ni_50.9_, the binding energy *E*_NC_ between hydrogen and surrounding B2 lattice atoms is somewhat higher than *E*_CG_ for coarse-grained specimens:*E*_CG_ = 31.4 ± 2.9 kJ/mol = 0.32 ± 0.03 eV,
*E*_NC_ = 44.9 ± 3.4 kJ/mol = 0.40 ± 0.03 eV.

These values of *E* are rather close to the binding energy of hydrogen atoms in stable octahedral B2 states of Ti_49.1_Ni_50.9_: 0.391 eV [26]. The fact that the high-temperature peak in thermal desorption spectra is associated with hydrogen localization at octahedral “titanium” B2 interstices is supported by experiments [35]. The experiments show that during electrolytic hydrogenation at low current density, no hydrides appear in surface layers of Ti_49.15_Ni_50.85_ (at%) alloy. Likely, a solid solution in the B2 lattice of specimens is formed by hydrogen atoms, and their thermal desorption spectra, thus, reveal a single high-temperature peak without any low-temperature one. Hence, it can be concluded that in our case of aging, hydrogen atoms diffuse from the near-surface hydride layer of Ti_49.1_Ni_50.9_ into the B2 structure via octahedral titanium interstices, forming a solid solution in the B2 phase. Because hydrogen atoms should first leave the less stable H2 states of TiNiH, no low-temperature peak appears in their thermal desorption spectra after aging. As for the H_1_ states of TiNiH, they are similar to stable octahedral titanium B2 states responsible for the high-temperature peak. According to ab initio calculations [26], as the hydrogen concentration in the B2 solid solution grows, the total B2 energy decreases steeply, while the total B19′ energy remains almost unchanged. As a result, the austenite and martensite energy difference tend to zero, and this provides B2 stabilization throughout the temperature range studied. Thus, the absence of a low-temperature peak and the increase in high-temperature peak intensity correlate with our experimental data on the effect of hydrogen redistribution in B2-phase Ti_49.1_Ni_50.9_ on its martensite transformation. Certainly, more complex nature of the high-temperature peak must not be ruled out. The large width of this peak and the high binding energy of hydrogen atoms and surrounding atoms in nanocrystalline specimens may suggest the possibility of two strongly overlapping peaks in its thermal desorption spectra. The first peak can be due to stable interstitial positions of hydrogen atoms, and the second to hydrogen localization in defect regions, e.g., grain boundaries. However, there is no way to reliably separate the high-temperature peak into components, and theoretical calculations are needed to clarify the thermal desorption of hydrogen in the tetragonal TiNiH structure.

## 4. Conclusions

Thus, our analysis shows that the effective hydrogen diffusion coefficient in nanocrystalline Ti_49.1_Ni_50.9_ is three times greater than its value in coarse-grained Ti_49.1_Ni_50.9_, which can be explained by the faster motion of hydrogen atoms along grain boundaries in the nanocrystalline material.

On aging at room temperature, both types of electrolytically hydrogenated B2-phase Ti_49.1_Ni_50.9_ reveal hydrogen redistribution from their near-surface layer deep into their bulk, occupation of octahedral interstitial sites by hydrogen atoms, and suppression of their B19′ transformations.

According to thermal desorption spectroscopy, two states of hydrogen atoms exist in freshly hydrogenated Ti_49.1_Ni_50.9_ irrespective of the grain size. After aging at room temperature, no low-energy states of hydrogen atoms are found in hydrogenated Ti_49.1_Ni_50.9_.

## Figures and Tables

**Figure 1 materials-12-03956-f001:**
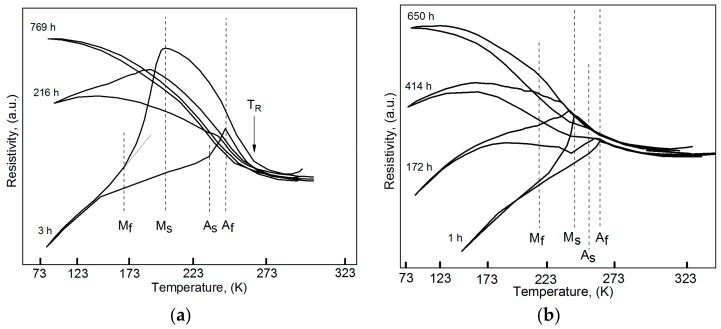
Temperature dependences of resistivity after hydrogenation and after further aging at room temperature for nanocrystalline (**a**) and coarse-grained (**b**) Ti_49.1_Ni_50.9_ [7].

**Figure 2 materials-12-03956-f002:**
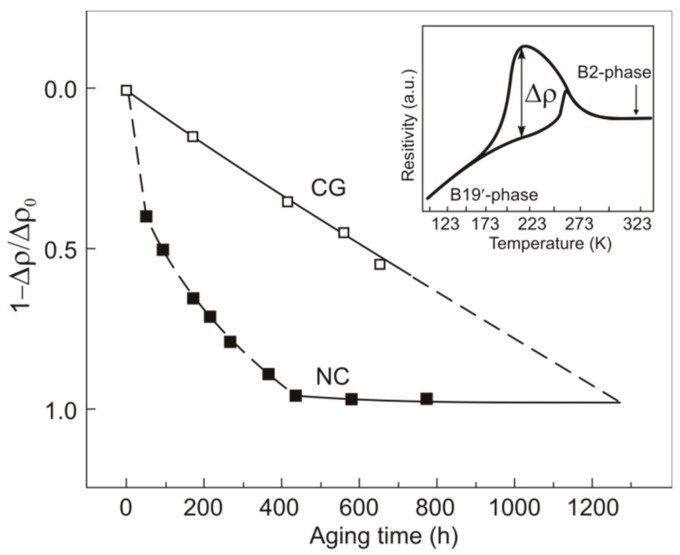
Time dependences of υ = (1 − Δρ/Δρ_0_) for hydrogenated Ti_49.1_Ni_50.9_ on aging: CG—coarse grained, NC—nanocrystalline, inset—diagram to estimate Δρ. Specimen diameter d = 1.1. mm.

**Figure 3 materials-12-03956-f003:**
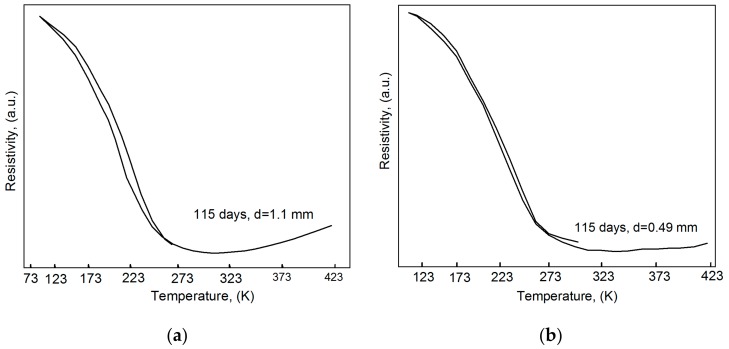
Resistivity–temperature curves of nanocrystalline Ti_49.1_Ni_50.9_ after hydrogenation and long-term aging. Specimen diameters: d = 1.1 mm (**a**) and d = 0.49 mm (**b**).

**Figure 4 materials-12-03956-f004:**
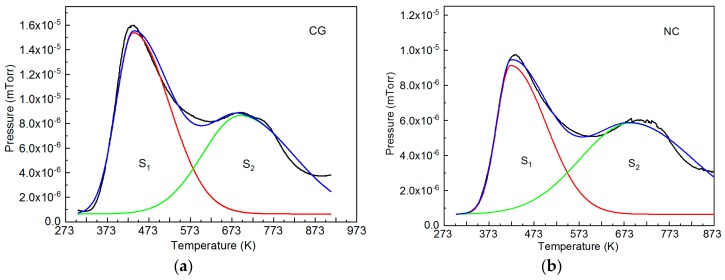
Thermal desorption spectra of hydrogen with separation into two components for freshly hydrogenated Ti_49.1_Ni_50.9_: CG—coarse-grained (**a**), NC—nanocrystalline (**b**). Heating rate 6 K/min. The relative areas of *S*_1_ and *S*_2_ components are presented in Table 3.

**Figure 5 materials-12-03956-f005:**
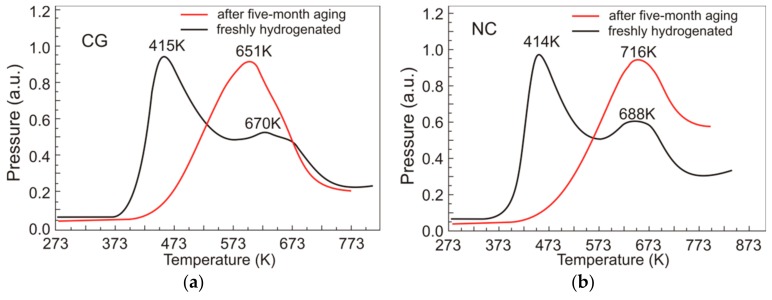
Thermal desorption spectra of hydrogen for Ti_49.1_Ni_50.9_ after hydrogenation and after five-month aging: CG—coarse-grained (**a**), NC—nanocrystalline (**b**). Heating rate 6 K/min.

**Figure 6 materials-12-03956-f006:**
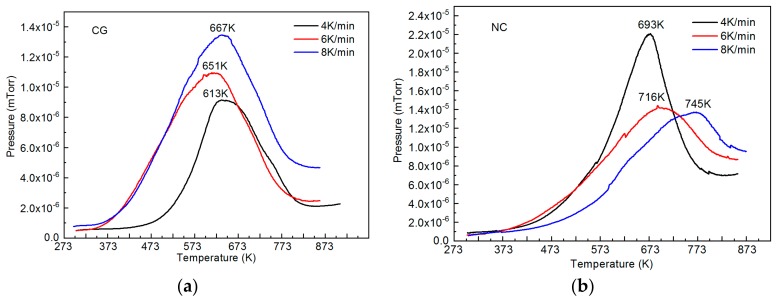
Thermal desorption spectra of hydrogen for Ti_49.1_Ni_50.9_ after five-month aging: CG—coarse-grained (**a**), NC—nanocrystalline (**b**). Heating rates 4, 6, and 8 K/min.

**Figure 7 materials-12-03956-f007:**
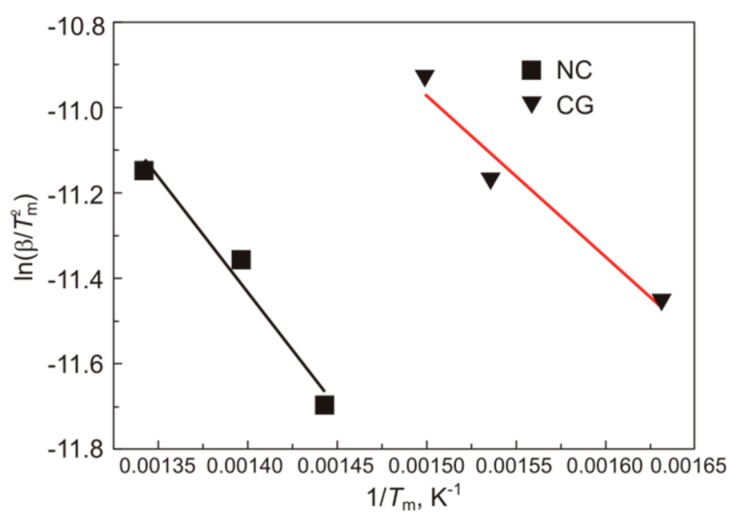
Dependences of $\ln(\beta /T_{m}^{2})$ on $1/T_{m}$ for nanocrystalline (NC) and coarse-grained (CG) Ti_49.1_Ni_50.9_.

**Table 1 materials-12-03956-t001:** Martensite transformation temperatures in Ti_49.1_Ni_50.9_ on cooling and heating *****.

Specimen Type and State	T_R_, K	M_s_, K	M_f_, K	A_s_, K	A_f_, K
Coarse-grained, initial [7]	-	241	221	252	263
Coarse-grained, aged for 1 h	-	239	220	253	261
Nanocrystalline, initial [7]	264	212	182	236	252
Nanocrystalline, aged for 3 h	264	210	165	234	254

***** T_R_ is the R-phase start temperature. M_s_, M_f_ are the martensite start and finish temperatures. A_s_, A_f_ are the austenite start and finish temperatures.

**Table 2 materials-12-03956-t002:** Hydrogen diffusion coefficients in B2-phase Ti_49.1_Ni_50.9_.

Specimen Type	*D*_H_, m^2^/s	Other Data for Comparison, *D*_H_, m^2^/s
Coarse-grained	6 × 10^−14^	2 × 10^−14^ [31]
		9 × 10^−15^ [16]
		5 × 10^−14^ [30]
Nanocrystalline	2 × 10^−13^	-

**Table 3 materials-12-03956-t003:** Relative areas *S*_1_, *S*_2_ beneath two Gaussians for Ti_49.1_Ni_50.9._

Specimen Type	*S*_1_, Rel. Units	*S*_2_, Rel. Units
Coarse-grained	0.53	0.47
Nanocrystalline	0.45	0.55

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
