# Peer review of "The Effect of Hydrogen on Martensite Transformations and the State of Hydrogen Atoms in Binary TiNi-Based Alloy with Different Grain Sizes"

_materials, 2019, doi:10.3390/ma12233956_

Round 1

Reviewer 1 Report

The work is a good theoretical and experimental investigation about the interaction of H with NiTi structure.

The results are well supported and discussed. i suggest only a better descritpion of the interesting resistivity curves, with precise indication of the point transition, for a better understanding of the reader. Even if there is a correct reference, the authors could consider some comparison with DSC analysis.

Author Response

Point 1: 
“I suggest only a better descritpion of the interesting resistivity curves, with precise indication of the point transition, for a better understanding of the reader.”

Response 1: The temperatures of martensitic transformations in Figure 1 are supplemented by.

Point 2: 
“Even if there is a correct reference, the authors could consider some comparison with DSC analysis.”

Response 2: Unfortunately, we cannot make a more detailed comparison with DSC measurement data. The results of the DSC analysis were obtained only for the initial nanocrystalline samples and hydrogenated samples after six months of aging published earlier in [7]. No the comparison of data, obtained by electrical resistivity measurements and DSC method was found in literature for the hydrogenated TiNi-based alloys.

Thank you for your attention to our work and useful comments.

Reviewer 2 Report

The paper reports on the effect of hydrogen diffusion on the phase transformations in NiTi shape memory alloys, and discusses the states of H atoms in the lattice and the diffusion coefficients in relation to the grain size.

The paper is clearly written and brings acceptably new results (while the resistivity measurements are rather incremental to [3], using the thermal desorption spectroscopy for analyzing the hydrogen states is novel for NiTi and enables new insights).

I recommend the paper to be accepted after minor revision, providing that the authors follow my below given recommendations (from which number 2 is essential).

1. Replace the phrase “know for sure” with something less informal in the first sentence of the introduction

2. Using the acronym PISS for the premartensitic state sound rather as a bad joke; the premartensite (precursor) effects are typically not considered as a new phase or structure, and saying that it appears in the tested samples as a part of some B2<–>PISS<–>B19 transition sequence is confusing. Putting a reference to [20], where no such state or acronym are introduced, makes things even worse. I know that the acronym ISS was used (without any additional explanation) in one figure in [3], but this does not justify introducing any PISS here. Anyway, the acronym appears explicitly vulgar and must be omitted. I strongly suggest calling the resistivity increase just “premartensite effect” or whatever, without introducing any, most probably non-existent, novel state.

3. The curves in Fig. 1 seem to be exactly the same as those in Fig. 5 in Ref [3]. If this is true, the reference should be given.

4. In Fig.7 and above, use T^2 (2 in superscript) for T squared, not just T2.

Author Response

Point 1: 
“Replace the phrase “know for sure” with something less informal in the first sentence of the introduction.”

Response 1: The phrase “know for sure” in introduction are replaced: “By now, it is known that hydrogenation…”

Point 2: 
“ Using the acronym PISS for the premartensitic state sound rather as a bad joke; the premartensite (precursor) effects are typically not considered as a new phase or structure, and saying that it appears in the tested samples as a part of some B2«PISS«B19¢ transition sequence is confusing. Putting a reference to [20], where no such state or acronym are introduced, makes things even worse. I know that the acronym ISS was used (without any additional explanation) in one figure in [3], but this does not justify introducing any PISS here. Anyway, the acronym appears explicitly vulgar and must be omitted. I strongly suggest calling the resistivity increase just “premartensite effect” or whatever, without introducing any, most probably non-existent, novel state.”

Response 2: PISS and ISS abbreviations are removed from the text. In revised variant of text we use only the phrase "premartensitic effect" to denote the increase in electrical resistivity when cooling coarse-grained and hydrogenated samples after aging at room temperature.

Point 3: 
“The curves in Fig. 1 seem to be exactly the same as those in Fig. 5 in Ref [3]. If this is true, the reference should be given.”

Response 3: The caption to figure 1 is supplemented by Ref. 7.

Point 4: 
“In Fig.7 and above, use T^2 (2 in superscript) for T squared, not just T2.”

Response 4: Formulas and designations in the text and in Figure 7 have been corrected:

ln(β/T2) replaced by ,

1/T replaced by .

Thank you for your attention to our work and useful comments.

Reviewer 3 Report

Dear Authors,

You have done a systematic job addressing the role of hydrogen charging (you call it hydrogenation) on the martensitic phase transformation (only thermally induced) of NiTi with superelastic composition. The grain  size effect is interesting and clearly shows the role of grain size as trap sites.

Please find below my comments.

Abstract:

1- Please remove the term "suppress". As I discuss it later on, defects can not "suppress" martensitic phase transformation. They only make it more difficult and change the mode of phase transformation from first-order (sudden) to second-order (continuous) i.e., martensitic phase transformation can still occur but in a different mode.

This is something that most people in SMA community do not pay attention. 

Results and discussion:

1- The first two paragraphs should be merged.

2- Second paragraph in my opinion is not a correct description of your resistivity measurements.  First of all, I can clearly see hysteresis in all the r(T) curves, indicating that THERMALY-INDUCED phase transformation is not fully suppressed. Even if hysteresis totally vanish, that does not mean phase transfromation is suppressed. As you see, r(T) curve increases in the hydrogenated specimen, if ideally there is no phase transformation in the specimen, resistivity should tend to zero in accordance with phonon theory. This clearly tells you that phase transformation is not suppressed and is still happening in your specimens. However, in the hydroganted specimen, the mode of phase transformation is not first-order. It occurs continuously.The continuous martensitic phase transfromation has been systematically addressed by Professor Q.P. Sun from Hong Kong University. They have published several papers in Acta Materialia in 2014 (showing continuous martensitic phase transformation using grain size topic and thermometry), 2015 (Using XRD), 2017 (using dilatometry and thermal expansion), and 2019 (using elastocaloric measurements). Please read Q.P. Sun works on this topic (martensitic phase transformation in defected and nanocrystalline specimens) and use it in the discussion of phase transformation in your doped specimens. You may refer to his articles or any other relevant article in the field of phase transformation in defected and doped crystals. You may also read Xiaobing Rens papers on strain glass in Physical Review Letters.

3-  You mention "which also follows from heat release and absorption measured by differential scanning calorimetry". You have published it...well...but its hard to get a very accurate DSC baseline... Not seeing a pick in DSC does not mean suppression of phase trasfromation. The transformation temperatures spreads over. Calorimetric measurements of Q.P Sun (Applied Physics Letters) proved that phase transformation still occurs. DSC is not just sensitive enough. r(T) measurements are better and you see narrow hysteresis.... Please revise. 

4- You mention "It is also seen from Figure 1 that after aging, the (T) curves of the hydrogenated specimens almost lose the hysteresis in the temperature range of martensite transformations.' Yes exactly ALMOST. Not completely, so it is not fully suppressed.

5- Can you perform new experiments on specimens with much smaller grain sizes, lets say 20 nm? 

6- You should cite this paper:

Effects of Hydrogen Charging on the Phase Transformation of Martensitic NiTi Shape Memory Alloy Wires (SMST Volume 3, Issue 4, pp 443–456)

I do not see any other flaws or deficiencies in your manuscript. You have discussed and explained all your observations accurately. 

Regards,

Reviewer

Author Response

Point 1: (Abstract)

“Please remove the term "suppress". As I discuss it later on, defects can not "suppress" martensitic phase transformation. They only make it more difficult and change the mode of phase transformation from first-order (sudden) to second-order (continuous) i.e., martensitic phase transformation can still occur but in a different mode.

This is something that most people in SMA community do not pay attention.”

Response 1: (Abstract)

Dear Reviewer! We would like to keep the term "suppress" in this version of our work. This term was used earlier to describe the effect of hydrogenation on the transformation into B19′ phase in most of the previous works. The possibility of changing the mode of the phase transition from first-order (sudden) to second-order (continuous) as a result of hydrogenated is not excluded. At present, however, we do not have any experimental results to support this assumption. At the same time, there are no obvious signs of transformation into B19′ phase (reduction of electrical resistance of samples at cooling below Ms) in hydrogenated and long-term aged (2760 h) samples. This will be presented in more detail in the responses to the remarks to the section "Results and discussion".

Point 1: (Section “Result and Discussion”)

“The first two paragraphs should be merged”

Response 1: (Section “Result and Discussion”)

The division into paragraphs 3.1 and 3.2 has been eliminated.

Point 2: (Section “Result and Discussion”)

“Second paragraph in my opinion is not a correct description of your resistivity measurements. First of all, I can clearly see hysteresis in all the ρ(T) curves, indicating that THERMALY-INDUCED phase transformation is not fully suppressed. Even if hysteresis totally vanish, that does not mean phase transfromation is suppressed. As you see, ρ(T) curve increases in the hydrogenated specimen, if ideally there is no phase transformation in the specimen, resistivity should tend to zero in accordance with phonon theory. This clearly tells you that phase transformation is not suppressed and is still happening in your specimens. However, in the hydroganted specimen, the mode of phase transformation is not first-order. It occurs continuously.The continuous martensitic phase transfromation has been systematically addressed by Professor Q.P. Sun from Hong Kong University. They have published several papers in Acta Materialia in 2014 (showing continuous martensitic phase transformation using grain size topic and thermometry), 2015 (Using XRD), 2017 (using dilatometry and thermal expansion), and 2019 (using elastocaloric measurements). Please read Q.P. Sun works on this topic (martensitic phase transformation in defected and nanocrystalline specimens) and use it in the discussion of phase transformation in your doped specimens. You may refer to his articles or any other relevant article in the field of phase transformation in defected and doped crystals. You may also read Xiaobing Rens papers on strain glass in Physical Review Letters.”

Response 2: (Section “Result and Discussion”)

The sufficiently long presence of small hysteresis of dependences ρ(T) at cooling and heating of aged samples of Ti49.1Ni50.9 (at.%) in work is connected only with slow saturation by hydrogen of the central zones of samples (i.e. enough long presence of inhomogeneous distribution of hydrogen). At the same time, we note that the volume fraction of the material experiencing the B2→B19′ or R→B19′ transformation decreases with increasing the duration of aging. This occurs in samples with a smaller diameter (0.49 mm) faster than in samples with a larger diameter (1.1 mm). The increase in electrical resistance during cooling is connected with the development of both the transformation of B2→R and the pretransition effects. The nature of these effects is not discussed in the paper. For this purpose it is necessary to study the structure of hydrogenated and aged samples in situ at cooling and heating by X-ray diffractometry and transmission electron microscopy methods. But these experiments are only planned. We do not exclude that the increase in the electrical resistance of the samples may also be due to the formation of "strain glass state", which was observed in the alloys Ti50-xNi50+x (at.%), Ti50Ni44.5Fe5.5 (at.%), and Ti49Ni51-xPdx [36-39]. These references are included in the text of the paper. We are well acquainted with the works of Prof. X. Ren on this subject, performed in the period from 2000 to 2018. However, experimental evidence of the development of "strain glass state" in hydrogenated TiNi-based alloys has not been found in the literature.

As we noted in Response 1 (Abstract), the possibility of changing the mode of phase transition from first-order to second-order as a result of hydrogenation is not excluded. But there are no experimental data confirming this assumption. We know the concept of Ahadi A. and Sun Q.P. about the changes in the mechanisms of phase transition from discontinuous to continuous in the nanocrystalline alloy Ti49.1Ni50.9 (at.%), occurring below a certain size of nanograins. The authors [I] have shown that the transition from discontinuous to continuous lattice changes occurs in the grain size range from 68 nm to 10 nm. At the same time, no indications of the continuous martensite transformation were found in nanocrystalline alloys Ti49.39Ni50.61 and Ti49.12Ni50.88 (at.%) with nanograins ranging in size from 17 nm to 8 nm [II, III]. The authors [II, III] assume that the continuous martensite transformation may be exist in nanocrystalline alloy with the grain size smaller than 8 nm.

In samples of our nanocrystalline alloy, the size of nanograins (85 nm) is larger than the critical size of nanograins in [I] (68 nm or less), in which the transition from discontinuous to continuous phase transformation was observed. Besides, in coarse-grained samples of our alloy (average grain size 10 μm) and samples with nanocrystalline structure the influence of aging on martensite transformations is similar. Thus, unfortunately, we have no reason to discuss our results in assuming of changes in the transition type from first-order to second-order, and changes in the transformation mechanism from discontinuous to continuous transformation.

References

I. Ahadi, A.; Sun, Q. Stress-induced nanoscale phase transition in superelastic NiTi by in situ X-ray diffraction. Acta Materialia. 2015, 90, 272-281.

II. Prokoshkin, S.; Dubinskiy, S.; Brailovski, V.; Korotitskiy, A.; Konopatskiy, A.; Sheremetyev, V.; Blinova, E. Nanostructures and stress-induced phase transformation mechanism in titanium nickelide annealed after moderate cold deformation. Materials Letters. 2017, 192, 111-114.

III. Prokoshkin, S.; Dubinskiy, S.; Korotitskiy, A.; Konopatskiy, A.; Sheremetyev, V.; Shchetinin, I.; Glezer, A.; Brailovski, V. Nanostructured features and stress-induced transformation mechanisms in extremely fine-grained titanium nickelide. J. Alloys Compd. 2019, 779, 667-685.

Point 3: (Section “Result and Discussion”)

“You mention "which also follows from heat release and absorption measured by differential scanning calorimetry". You have published it...well...but its hard to get a very accurate DSC baseline... Not seeing a pick in DSC does not mean suppression of phase trasfromation. The transformation temperatures spreads over. Calorimetric measurements of Q.P Sun (Applied Physics Letters) proved that phase transformation still occurs. DSC is not just sensitive enough. r(T) measurements are better and you see narrow hysteresis.... Please revise.”

Response 3: (Section “Result and Discussion”)

The authors agree with the reviewer's comment. We excluded comparison with DSC analysis results from revised version of manuscript. Systematic studies of the effect of aging duration on heat release and absorption have not been conducted in our work. The results of DSC mesurements performed on nanocrystalline samples in their initial state and after six-month aging were published in [7].

 Point 4: (Section “Result and Discussion”)

“You mention “It is also seen from Figure 1 that after aging, the r(T) curves of the hydrogenated specimens almost lose the hysteresis in the temperature range of martensite transformations.” Yes exactly ALMOST. Not completely, so it is not fully suppressed.”

Response 4: (Section “Result and Discussion”)

We agree with the reviewer's comment. Corresponding changes have been made to the text. In the first approximation, we can assume that the transformation to B19¢ phase is practically suppressed only in nanocrystalline hydrogenated samples with a diameter of 0.49 mm (Figure 3b).

Point 5: (Section “Result and Discussion”)

“Can you perform new experiments on specimens with much smaller grain sizes, lets say 20 nm?”

Response 5: (Section “Result and Discussion”)

Such experiments will be carried out.

Point 6: (Section “Result and Discussion”)

“You should cite this paper:

Effects of Hydrogen Charging on the Phase Transformation of Martensitic NiTi Shape Memory Alloy Wires (SMST Volume 3, Issue 4, pp 443–456)”

Response 6: Thank you for recommending this very interesting work. We have used the results of this work in the text of the article. Unfortunately, the electric resistivity curves in temperature range of martensite transformations and the thermal desorption spectra of hydrogen was not studied, in the article by Snir, Y.; Carl, M.; Ley, N.A.; Young, L.M. "Effects of Hydrogen Charging on the Phase Transformation of Martensitic NiTi Shape Memory Alloy Wires".

Thank you for your attention to our work and useful comments.

Reviewer 4 Report

In the current manuscript, the authors have deeply analyzed the diffusion of hydrogen and its effect on the martensitic transformation of Ti-Ni wires. The explanation given by the authors about the suppression of the martensite transformation is plausible and coherent with the data, which could justify the publication of the work. However, the section 3.1 is partially published in reference 3 of the manuscript. Therefore, I must suggest the authors to deeply review the structure of the work. I would like also to mention some weakness that could be improved in a future version:

In the introduction, I recommend to the authors to motivate the study of hydrogenation of Ti-Ni alloys. Add an explanation (in text or in the caption) of the meaning of each curve in Figure 4. I strongly recommend including the uncertainties to the binding energies to determine how different are ECG and ENC.

Author Response

Point 1: 
“However, the section 3.1 is partially published in reference 3 of the manuscript. Therefore, I must suggest the authors to deeply review the structure of the work. I would like also to mention some weakness that could be improved in a future version.”

Response 1: The structure of the article has been redesigned. Sections 3.1 and 3.2 are merged. Descriptions of the results (in particular, dependences ρ(T) in hydrogenated samples compared to their description given in Ref. 7) are supplemented.

Point 2: 
“In the introduction, I recommend to the authors to motivate the study of hydrogenation of Ti-Ni alloys. Add an explanation (in text or in the caption) of the meaning of each curve in Figure 4.”

Response 2: The motivations of studies of hydrogenated TiNi-based alloys are introduced in section “Introduction”. The caption to Figure 4 is supplemented.

Point 3: 
“I strongly recommend including the uncertainties to the binding energies to determine how different are ECG and ENC.”

Response 3: The uncertainties of ECG and ENC are presented.

Thank you for your attention to our work and useful comments.

Round 2

Reviewer 4 Report

In view of the changes made by the authors in the new version of the manuscript, I am pleased to recommend the acceptance of the work for publication.